# SELF-DISTILLATION FOR DIFFUSION

## ABSTRACT

In recent years, diffusion models have demonstrated powerful generative capabilities. As they continue to grow in both ability and complexity, performance optimization becomes more relevant. Knowledge Distillation (KD), where the output from a pre-trained teacher model is used to train a smaller student model, has been shown to greatly reduce the number of network evaluations required, while retaining comparable image sample quality. KD is especially useful in diffusion, because it can be used not only to distill a large model into a small one, but also to distill a large number of denoising *iterations* into a small one. Here, we show that a form of *self-distillation*—training a subnetwork to mimic the output of the larger network, effectively distilling a network into itself—can improve distillation in diffusion models. We show first that when a pre-trained teacher model is distilled to a student network, we can turn this into a self-distillation procedure by unifying the teacher and the student. Our results indicate that this leads to faster convergence for a competitive sample quality. Additionally, we show in small-scale experiments that when diffusion models are trained from scratch, adding a self-distillation term to the loss can, in specific cases, help the model to convergence, producing high-quality samples more quickly.

## 1    INTRODUCTION

A major drawback of image-generating diffusion models (Sohl-Dickstein et al., 2015; Song & Ermon) is the gradual denoising process, at times requiring up to 1000 denoising steps in order to produce results which are visually pleasing and/or acceptable interpretations of the desired final outcome (Sohl-Dickstein et al., 2015; Ho et al., 2020). Optimizations, such as training on latent image representations and using implicit scheduling, known as denoising diffusion implicit models (DDIM), have reduced the number of steps needed to generate decent samples to the range of 20-50 (Song et al., 2021). Despite these advances, diffusion models still require substantially more computational resources at inference time than Generative Adversarial Networks (GANs), whose execution is comparable in complexity to a single iteration of a diffusion model (Sohl-Dickstein et al., 2015; Ho et al., 2020). Recently, knowledge Distillation (KD) techniques have proven effective in further reducing the required DDIM steps, in some cases even achieving good results in a single forward pass (Salimans & Ho, 2022; Meng et al., 2023; Luhman & Luhman, 2021).

Current distillation techniques for diffusion models involve a form of teacher/student distillation, where an original pre-trained model is frozen, to be used as a teacher, and is *copied*, with the copy to be used as a student model. ==The student is trained to mimic the teacher, but in fewer iterations, so that in time the student converges to replicate the teacher's sample quality at fewer sampling steps.== Either the student's single DDIM step is directly compared against the teacher's final output (Luhman & Luhman, 2021), which involves a slow training process requiring many evaluations per parameter update, or the student's single-step output is compared against two steps of the teacher model, progressively reducing the total DDIM steps in halves (Salimans & Ho, 2022). This results in three steps being taken per update of the student's weights (Salimans & Ho, 2022; Meng et al., 2023). For both these existing methods, the teacher model's parameters remain static during each iteration, providing the same quality of information for the student to absorb (Song et al., 2021).

A different approach to distillation is *self-distillation*[1] (Zhang et al., 2022; 2019), where during training, a model (like a CNN or a single-stack LLM) is given two outputs: one from all layers in the model, and one from only the first $k$ layers. Then as the model trains on a standard loss, an additional loss term is added to induce the second output to mimic the first. The idea is that whatever the whole

network has learned is distilled into the lower layers, freeing up the higher layers. In this setting there is no need for a two-phase training: as soon as the network as a whole learns something, it is immediately distilled into the subnetwork.

We propose to apply self-distillation to diffusion models, resulting in what we term *direct self-distillation* (DSD). In DSD, we do not separate the teacher and student models and instead, in the spirit of self-distillation, distill a model *into itself* during training.

Specifically, we train sequentially, taking a full DDIM sample from the model. For every two steps of sampling, receiving intermediate samples $\mathbf{z}_{t-1}, \mathbf{z}_{t-2}$, we then take the current prediction for the fully denoised final sampling step $x_0(\mathbf{z}_{t-1})$, with the subsequent step's predicted $x_0(\mathbf{z}_{t-2})$ as a target, and minimize the squared distance between them (computing gradients only over the prediction).

Compared to existing methods, there are three key differences. First, we distill the model into itself, instead of keeping the teacher and student separate. Second, we do not use training data, instead relying only on the samples from the model itself. This simplifies training, removing data loading from the process entirely. Finally, we train *sequentially*: we perform distillation steps for every other iteration of a full sample. This contrasts with previous methods where a random point along the sampling trajectory was chosen for each distillation step.

The result is a diffusion model that directly generates high-quality output in a low number of steps. We show that a large diffusion model (specifically, a pretrained conditional ImageNet 256x256 model with ∼400M parameters as well as pretrained LSUN Bedroom and CelebA unconditional models with ∼274M parameters each, at 128x128 and 256x256 resolutions respectively (Rombach et al., 2022)) can be distilled down to two iterations, using DSD. Similarly, We follow the experiments from (Yang et al.), and show that if the teacher/student distillation used there is replaced by DSD, competitive results may be achieved with as much as an order of magnitude fewer parameter updates.

Then, to test whether, in principle at least, self-distillation may be applied when training a model from scratch, we introduce a version of DSD for this purpose, and show that it performs well on small scale image generation tasks.[2] Taken together with our other results, this suggests that when training image diffusion models at scale, self-distillation may to help reduce the number of iterations required as part of training, with no need for a two-stage process.

## 2   RELATED WORK

Diffusion models were introduced in Sohl-Dickstein et al. (2015) and shown to perform at scale in Ho et al. (2020). Despite advances like DDIMs Song et al. (2021) to reduce the number of iterations required, the sampling time still remains one of the main bottlenecks.

Knowledge distillation (Buciluǎ et al., 2006; Hinton et al., 2015; Gou et al., 2021) is a promising approach to reduce the required iterations to (close to) 1. This was first done by Luhman & Luhman, distilling directly into a single network using a fully denoised image as targets. Salimans & Ho (2022) introduced *Progressive distillation*, showing that gradually reducing the number of steps taken by both the teacher and the student leads to better results whilst introducing intermediate denoised images as targets, improving efficiency. Meng et al. (2023) extended this approach to conditional diffusion models.

One application of knowledge distillation is that large networks may be distilled into smaller networks with relatively little loss of performance (Sanh et al.). This leads naturally to the idea of *self-distillation* (Zhang et al.; 2021), where the output of a large network is distilled into a subnetwork during training.[3] This often improves performance at very little extra computational cost. Self-distillation has successfully been applied in contexts such as graph learning (Li et al., 2021), object detection (Zheng et al., 2020) and regularizing classification (Yun et al., 2020).

---

[1]The phrase *self-distillation* is also used to refer to teacher/student distillation with identical architectures. Here, it only refers to the method described, where the student is in some sense a sub-network of the teacher.

[2]Resource limitations preclude us from testing this approach on the scale of state-of-the-art models.

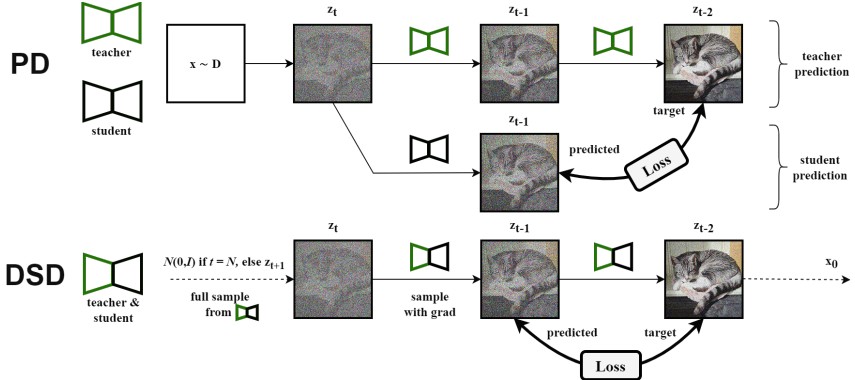

Figure 1: Diagram outlining the processes and distinctions between Progressive Distillation (from Salimans & Ho) (PD) and Direct Self-Distillation (DSD).

A related, but distinct approach is to use a trained model to generate several outputs, and to combine these into a superior training label on which the whole model is then refined. This may be referred to as *self-training* in large language models Huang et al. (2022) and a *policy-improvement operator* in reinforcement learning Sutton & Barto (2018); Silver et al. (2017).

## 3 PROGRESSIVE DISTILLATION (PD)

Progressive Distillation (PD) aims to retain sampling quality whilst progressively halving DDIM steps taken. This method iteratively refines a diffusion model by reducing the number of sampling steps, utilizing a teacher-student dynamic. The process commences with a standard-trained teacher diffusion model, which then imparts its characteristics to a student model.

In each iteration of PD, the student model is initialized as a clone of the teacher model, inheriting both parameters and model definition. The training involves sampling from a data-set, adding noise to it to form noisy sample $z_t$, and then applying the student model to this data. Unlike conventional training, the PD approach modifies the target for the denoising task. Instead of the original data $x$, the student model denoises towards a newly defined target $\tilde{x}$, calculated as:

$$\tilde{x}(z_t) = 2 \text{ DDIM steps with teacher model from } z_t \text{ to } z_{t-1/N}$$

where $N$ is the number of student sampling steps. This process sees three DDIM steps per student model parameter update. The refined teacher target, specific to PD, allows the student model to make more precise predictions, enhancing sampling efficiency. After training a student model with $N$ steps, the process is repeated with $N/2$ steps, where the student then serves as the new teacher. This iterative cycle, detailed in Algorithm 1, continues until a desired efficiency is achieved, significantly reducing sampling time while maintaining output quality.

## 4 DIRECT SELF-DISTILLATION

We will refer the general family of approaches that take a pre-trained model (the teacher) along with an image data-set, and distill it into a separate, new model (the student) as **progressive distillation (PD)** techniques. By contrast, we aim to distill a single model into itself, removing the both the need of a static teacher model or image data-set. That means that the distillation step at any point in

---

[3]When we apply this to distillation, we take two iterations of the UNet model, and distill the result back into the first. Therefore we don't quite distill into a smaller sub-network, but essentially do distill the output of a larger *computation graph* into a smaller sub-graph.

training attempts to teach the pre-trained model to do in $N$ steps what it itself currently does in $2N$ steps. Variants of this approach will be labeled as forms of **direct self-distillation (DSD)**.

## 4.1 FINE-TUNING DSD

Let $\mathbf{x}_0 \sim \mathcal{D}$ represent a sample from the dataset. Diffusion works by generating a sequence of $T$ latent variables $\mathbf{z}_1, \ldots \mathbf{z}_T$, by progressively adding noise to the data, so that $\mathbf{z}_T \sim N(\mathbf{0}, \boldsymbol{I})$. A UNet model[4] $\hat{\boldsymbol{x}}_\theta$ is then trained to reverse this process. In most diffusion models $\hat{\boldsymbol{x}}_\theta(\mathbf{z}_t)$ predicts the noise vector $\boldsymbol{\epsilon}_t$ for which $\mathbf{z}_t = \mathbf{x}_0 + \boldsymbol{\epsilon}_t$. From this, we can predict both $\mathbf{x}_0$ and $\mathbf{z}_{t-1}$. Sampling is then performed by starting with a sample $\mathbf{z}_T \sim N(\mathbf{0}, \boldsymbol{I})$ and progressively denoising it.

In direct self distillation, we are given a pre-trained model $\hat{\boldsymbol{x}}_\theta$. We perform a full DDIM sample of $N$ iterations. Note that when we sample from $\hat{\boldsymbol{x}}_\theta$, $t$ indexes backwards from $N$ to $0$.

We compute two steps of DDIM for every step of self-distillation. Starting at $\mathbf{z}_t$, this produces $\mathbf{z}_{t-1}$ and $\mathbf{z}_{t-2}$.

We then update according to the loss function $w(\lambda_t) \cdot \|\mathbf{x}_{t-1} - \mathbf{x}_{t-2}\|^2$ where $w(\lambda_t)$ represents a weighting function. We back-propagate only over the computation of $\mathbf{z}_{t-1}$, detaching the computation of $\mathbf{z}_{t-2}$. This ensures that in the above loss, $\mathbf{x}_{t-1}$ functions as a prediction, and $\mathbf{x}_{t-2}$ functions as a target, with the network learning to adapt the former to the latter. See Figure 1 for a diagrammatic comparison between DSD and PD.

As a weighting $w(\lambda_t)$, we use the Truncated Signal to Noise Ratio (SNR), which allows for the distillation to put more emphasis on updating model parameters given higher SNR values aiming to improve image sharpness. $w$ in turn truncates weighting during early DDIM steps with low SNR, which becomes increasingly important when DSD progresses towards fewer DDIM steps with higher noise values.

Below we outline the DSD algorithm, together with the progressive distillation (PD) algorithm for comparison. Note that in DSD $\theta$ represents the parameters of the *given model* which is fine-tuned directly.

### 4.1.1 SCHEDULING $N$

In PD, the student is trained until convergence, after which the number of sampling steps $N$ is halved and the old student becomes the new teacher.

In DSD, we do not require that training converges.[5] Instead, we define three different schedules for how $N$ should decrease during training, and how many samples we take for each $N$.[6]

Experimentally, we find a good approach is to keep the total parameter updates seen for each depth $N_i$ approximately constant. If, for instance, we decay $N_i$ linearly, but keep the number of samples for each $N_i$ the same, more parameter updates are performed for the higher $N_i$, since we update once for every two sampling steps. Instead, we adjust the number of samples taken, referred to as $U_i$, to keep the number of parameter updates for each $N_i$ roughly equal.

**Naive DSD (DSDN)**   In DSDN, $N_i$ and $U_i$ remain constant throughout distillation. This approach does not ultimately reduce the number of denoising steps required for generation.

Note however, that we still distill later denoising steps into earlier ones, so we can still expect to see high-quality output after fewer steps than in the original model, if training is successful.

---

[4]The model also receives $t$ as a parameter, but this is omitted to simplify the notation.

[5]If convergence-based scheduling is preferred, intermittent FID/IS score calculation can provide such metrics in the absence of a converging loss function.

[6]Since we do not require a new model to be copied when we change $N$, we are more flexible in reducing $N$ gradually.

| **Algorithm 1** PD (from (Salimans & Ho, 2022)) | **Algorithm 2** Fine-tuning DSD |
|---|---|
| **Require:** Trained teacher model $\hat{\boldsymbol{x}}_\eta(\boldsymbol{z}_t)$ 
 **Require:** Data set $\mathcal{D}$ 
 **Require:** Loss weight function $w()$ 
 **Require:** Sampling steps $N$ | **Require:** A trained model $\hat{\boldsymbol{x}}_\theta(\boldsymbol{z}_t)$ 

 **Require:** Loss weight function $w()$ 
 **Require:** Schedule of # of updates $U$, and steps $N$ |

**Algorithm 1:**

> **for** $K$ iterations **do**
>   $\theta \leftarrow \eta$       ▷ Init student from teacher
>   **while** not converged **do**
>     $\mathbf{x} \sim \mathcal{D}$
>     $t = i/N, \;\; i \sim Cat[1, 2, \ldots, N]$
>     $\epsilon \sim N(0, I)$
>     $\mathbf{z}_t = \alpha_t \mathbf{x} + \sigma_t \epsilon$
>     # 2 steps of DDIM
>     $t' = t - 0.5/N, \;\; t'' = t - 1/N$
>     $\mathbf{z}_{t'} = \alpha_{t'} \hat{\boldsymbol{x}}_\eta(\boldsymbol{z}_t) + \frac{\sigma_{t'}}{\sigma_t}(\mathbf{z}_t - \alpha_t \hat{\boldsymbol{x}}_\eta(\boldsymbol{z}_t))$
>     $\mathbf{z}_{t''} = \alpha_{t''} \hat{\boldsymbol{x}}_\eta(\boldsymbol{z}_{t'}) + \frac{\sigma_{t''}}{\sigma_{t'}}(\mathbf{z}_{t'} - \alpha_{t'} \hat{\boldsymbol{x}}_\eta(\boldsymbol{z}_{t'}))$
>     $\tilde{\mathbf{x}} = \frac{\mathbf{z}_{t''} - (\sigma_{t''}/\sigma_t)\mathbf{z}_t}{\alpha_{t''} - (\sigma_{t''}/\sigma_t)\alpha_t}$   ▷ Teacher $\hat{\mathbf{x}}$ target
>     $\lambda_t = \log[\alpha_t^2/\sigma_t^2]$
>     $L_\theta = w(\lambda_t)\|\tilde{\mathbf{x}} - \hat{\boldsymbol{x}}_\theta(\boldsymbol{z}_t)\|^2$
>     $\theta \leftarrow \theta - \gamma \nabla_\theta L_\theta$
>   **end while**
>   $\eta \leftarrow \theta$      ▷ Student becomes next teacher
>   $N \leftarrow N/2$      ▷ Halve # of sampling steps
> **end for**

**Algorithm 2:**

> **for** $U_i$ updates **do**
>   **for** $t \in N_i, N_i - 2, N_i - 4, \ldots, 2$ **do**
>     $\mathbf{z}_t = N(0, I)$ **if** $t = N$ **else** $\mathbf{z}_{t''}$
>     # 2 steps of DDIM
>     $t' = t - 1, \;\; t'' = t - 2$
>     $\mathbf{z}_{t'} = \alpha_{t'} \hat{\boldsymbol{x}}_\theta(\boldsymbol{z}_t) + \frac{\sigma_{t'}}{\sigma_t}(\mathbf{z}_t - \alpha_t \hat{\boldsymbol{x}}_\theta(\boldsymbol{z}_t))$
>     $\mathbf{z}_{t''} = \alpha_{t''} \hat{\boldsymbol{x}}_\theta(\boldsymbol{z}_{t'}) + \frac{\sigma_{t''}}{\sigma_{t'}}(\mathbf{z}_{t'} - \alpha_{t'} \hat{\boldsymbol{x}}_\theta(\boldsymbol{z}_{t'}))$
>     $\mathbf{x}' = \frac{\mathbf{z}_{t'} - (\sigma_{t'}/\sigma_t)\mathbf{z}_t}{\alpha_{t'} - (\sigma_{t'}/\sigma_t)\alpha_t}$   ▷ predicted $\hat{\mathbf{x}}$
>     $\mathbf{x}'' = \frac{\mathbf{z}_{t''} - (\sigma_{t'}/\sigma_t)\mathbf{z}_t}{\alpha_{t''} - (\sigma_{t'}/\sigma_t)\alpha_t}$   ▷ target $\hat{\mathbf{x}}$, detached
>     $\lambda_t = \log[\alpha_t^2/(1 - \alpha_t^2)]$
>     $L_\theta = w(\lambda_t)\|\mathbf{x}' - \mathbf{x}''\|^2$
>     $\theta \leftarrow \theta - \gamma \nabla_\theta L_\theta$
>   **end for**
> **end for**

**Iterative DSD (DSDI)** In DSDI, we implement a similar scheduling approach to that of progressive distillation, where the total steps are progressively halved during the distillation process.

We first set the total number of parameter updates $P$. We then have $N_{i+1} = N_i/2$ and $U_i = \frac{P}{|N|}\frac{1}{N_i}$, with $|N|$ being the total number of different step-size depths $N_i$ we use.

Additionally, we introduce **teacher-student distillation (TSD)** with identical scheduling to DSDI, which similarly forgoes the sampling from an existing data-set, yet differs from DSD in that we instead mimic PD and introduce a separate teacher model to produce the distillation targets.

**Gradual DSD (DSDGL)** In gradual DSD, we decay the number of learning steps by two steps per iteration, so that $N_{i+1} = N_i - 2$. The formula for $U_i$ is the same as before.

## 4.2 TRAINING DSD

If self-distillation works for finetuning an already-trained model, it is reasonable to ask whether we might also apply it during the original training phase of the model. In this setting, we are training from data, and we are not necessarily sampling from the model, so we adapt our method to minimize the amount of overhead that the inclusion of self-distillation requires.

Given that direct self-distillation does not require any pre-trained teacher model, it becomes possible for a diffusion model to distill into itself during the training phase. During such a process the distillation loss, calculated as the error between two successive DDIM steps as depicted in 1, can either be used to update the diffusion model's parameters, or alternatively incorporated in combination with the train loss during each step.

The training loss is the squared difference between the predicted $\hat{\mathbf{v}}_t$ and the true $\mathbf{v}_t$, following the v-prediction algorithm as described in (Ho et al.). The total loss is then supplemented by the distillation loss between the predicted $\mathbf{x}'$ and the predicted $\mathbf{x}''$, similar to that of Algorithm 2:

$$L = \alpha \cdot [\hat{\mathbf{v}}_t - \mathbf{v}_t]^2 + \beta \cdot [\mathbf{x}' - \mathbf{x}'']^2 \qquad (1)$$

where $\alpha$ and $\beta$ are hyper-parameters used to balance the main loss and the distillation loss respectively. Due to $\hat{\mathbf{z}}_t$ being a component in both the calculation of $\hat{\mathbf{v}}_t$ and $\mathbf{x}'$, this only needs to be calculated once with a single computation graph.

## 5 RESULTS

**Evaluation and training details**  To compare these different methods, we use the Fréchet Inception Distance (FID) and Inception Score (IS) to evaluate image sampling performance at various DDIM steps. For unconditional image generation, we use the models from Rombach et al. (2022) pre-trained on the CelebA-HQ $256^2$ dataset (Karras et al., 2018) and LSUN Bedroom (Yu et al., 2015),[7] as well as the conditional model from the same source trained on the ImageNet-$256^2$ dataset (Deng et al., 2009). We compare three scheduling approaches for DSD to values reported in the relevant PD literature, in addition to TSD.

During the distillation process, we set the guidance scale to be randomly sampled from $w = [1.0, 3.0]$ and all distillation attempts for the conditional and unconditional models receive $4\,000$ and $5\,000$ total parameter updates ($P$), respectively. The hyperparameters used for both the conditional- and unconditional models are shown in Table 3, in the appendix. In previous research on the distillation of conditional diffusion models by (Meng et al., 2023), Progressive Distillation required re-training of the original model to predict $\mathbf{x}_0$ without the use of a separate guidance forward pass, as running both passes caused divergence during the distillation process. Our approach does not involve this procedure, as our goal is not to distill down to as few sampling steps as possible, but rather to compare DSD to Progressive Distillation while reducing the required model updates for similar image quality.

The learning rates for all three models were kept identical between different distillation procedures to ensure a fair comparison between the DSD/TSD implementations and PD. More specifically, we optimized the learning for TSD and then applied these values to DSD without further optimization, to achieve a conservative estimate of the relative benefit of DSD over PD, given that TSD more closely resembles PD.

The implementation of progressive distillation by Ho et al. saw the learning rate reduced to 0 during each halving of total DDIM steps. Given our lower number of total parameter updates, we decided to implement a cosine annealing schedule for the learning rate, gradually decreasing to $10\%$ of the original learning rate over the duration of each distillation procedure. This was found to improve the distillation stability of TSD, and therefore by extension DSDI, which both see fewer DDIM steps as the distillation progresses. This annealing schedule was then applied to all DSD methods.

### 5.1 FINE-TUNING DSD

Table 1 shows the FID scores for the original unconditional CelebA-HQ and LSUN Bedroom models, as well as the FID/IS scores for the conditional ImageNet-256 model.

**Unconditional models**  Distillation results in terms of FID scores of the unconditional CelebA-HQ and LSUN Bedroom models are shown in Table 2, with the undistilled models' performances detailed in Table 1. A visual comparison between the different DDIM steps for CelebA-HQ is given in Appendix 2. Given the absence of image classes for either dataset, only FID scores were used to compare image fidelity between different distillation types and the number of DDIM steps. For CelebA-HQ, for which at the time of writing no PD counterpart was found, both 2- and 4 step DSDI show a minor lead over other self-distillation types, with DSDN demonstrating a slight improvement at 8 steps. TSD shows what performance may be expected when performing *indirect* self-distillation

---

[7]The original pretrained LSUN Bedroom model was configured to produce latent images at 64x64, before being upsampled within the pixel-space to 256x256, similar to the other models. We altered the configuration to match the output of 128x128 for comparison against results obtained from the Progressive Distillation implementation by Salimans & Ho.

| | FID(↓)/IS(↑) | | | | | |
|---|---|---|---|---|---|---|
| | $S=2$ | $S=4$ | $S=8$ | $S=16$ | $S=32$ | $S=64$ |
| ImageNet-256 | 142.95/3.70 | 32.66/30.72 | 10.27/151.33 | 9.31/189.64 | 7.72/198.97 | 6.90/201.64 |
| CelebA-HQ | 217.95/- | 98.77/- | 51.53/- | 31.86/- | 25.18/- | 22.80/- |
| LSUN Bedroom | 303.41 /- | 78.64 /- | 23.48 /- | 12.30 /- | 10.70 /- | 9.79 /- |

Table 1: FID scores for the pre-trained CelebA-HQ and LSUN Bedroom models and FID/IS scores for the Conditional ImageNet-256 model, prior to distillation, at *2, 4, 8, 16, 32,* and *64* DDIM steps, calculated over 5K sample images per combination of model and sampling-steps (S).

| Model | Conditional DDIM - FID(↓)/IS(↑) | | | | |
|---|---|---|---|---|---|
| | PD | TSD (ours) | DSDI (ours) | DSDN (ours) | DSDGL (ours) |
| ImageNet-256 2-step (5K) | **14.55 / 120.00** (52K*) | 129.54 / 4.60 | 119.98 / 5.20 | 100.26 / 7.98 | 132.03 / 4.38 |
| ImageNet-256 4-step (5K) | 13.19 / 126.36 (32K*) | 20.21 / 61.33 | 15.15 / 87.85 | **10.27 / 126.61** | 20.01 / 82.70 |
| ImageNet-256 8-step (5K) | 12.72 / 128.63 (12K*) | **9.51 / 180.49** | 9.78 / 184.24 | 13.69 / **198.43** | 14.46 / 140.68 |

| Model | Unconditional DDIM - FID(↓) | | | | |
|---|---|---|---|---|---|
| CelebA-HQ 2-step (4K) | - | 151.01 | **136.45** | 137.38 | 145.61 |
| CelebA-HQ 4-step (4K) | - | 69.12 | **41.46** | 45.78 | 52.40 |
| CelebA-HQ 8-step (4K) | - | 28.92 | 19.51 | **19.28** | 24.23 |
| Lsun Bedroom 2-step (4K) | **6.70** (400K**) | 183.76 | 141.87 | **104.48** | 142.24 |
| Lsun Bedroom 4-step (4K) | **3.53** (450K**) | 78.69 | 43.56 | **30.50** | 38.91 |
| Lsun Bedroom 8-step (4K) | **2.31** (550K**) | 23.59 | 17.87 | **11.20** | 28.01 |

Table 2: FID scores at *2, 4* and *8* DDIM steps for the distilled unconditional CelebA-HQ and LSUN Bedroom pretrained models (calculated after 4 000 parameter updates) and FID/IS scores for the distilled Conditional pretrained ImageNet-256 model (calculated after 5 000 parameter updates). The Inception Scores were calculated using 5 000 generated samples for each combination of the model, step size, and distillation procedure, using 30 000 samples for the FID calculation. *PD* shows baseline results of previous Progressive Distillation techniques, with **\*** indicating results and number of model parameter updates from figures in Meng et al., compared against 5 000 samples. **\*\*** Indicates results and the amount of model parameter updates similarly obtained from figures in Salimans & Ho.

method including a static teacher. TSD's direct-distillation counterpart, DSDI, outperforms TSD by a large margin at virtually all sampling durations tested.

Interestingly, for the LSUN Bedroom model, DSDN outperforms other direct self-distillation methods by quite a large margin, although the very low number of sampling steps do not show image fidelity quite like that of the Progressive Distillation baseline gathered from Salimans & Ho. Regardless, a great improvement to image quality is gained at these steps with just 4 000 parameter updates, which requires far fewer steps than PD at around 400 to 500 thousand updates. It should however be noted that our implementation of distillation, particularly self-distillation, may be introducing variance to the pixel-distributions of the model's generated images. This is due to the sampling of distillation targets from the model's own output, compared to existing methods which sample real images before adding Gaussian noise. Inception Scores are therefore more representative of image quality and diversity, although unavailable when sampled without conditional classes.

**Conditional models** For the distilled class-conditional ImageNet-256 models, Table 2 shows the results of the distilled models in terms of FID and IS scores. Regarding IS, DSDN outperforms all other DSD methods, suggesting that DSDN is able to improve distinct image features faster. In Figure 2 we see a comparison given the same initial noisy latent $z_0$, which shows how DSDN seems to outperform the other distilled models after both 4 and 8 steps in subjective image quality as well. At 4 steps the outline of the four classes appears to have more detail and clarity, plus improved contrast and colour saturation. At 8 steps, some novel image structures are introduced, where for example the caudal fin (vertical tail fin) of the *great white shark* sample is defined only in the DSDN model, even improving upon the image coherence of the respective original 64-step model's image. Such examples of DSDN seemingly surpassing the original full-step model were common during sample generation.

Compared to the existing Progressive Distillation method for conditional models by Meng et al., we are outperformed at very low DDIM steps. At this stage they performed between 12-52 thousand parameter updates, while our results were obtained after just 5 000. Still, especially DSDN, sees improved image quality at 4 and 8 DDIM steps despite the relatively short training period. Interestingly, both the FID and IS scores as detailed by Meng et al. don't improve much as the number of steps increase.

We note the large difference between the minimum FID scores obtained by the baseline PD papers compared to our methods. It is possible that our decision to perform distillation up to a maximum of 64 DDIM steps may have contributed to increased deviation from the data-sets' pixel distributions. This could be explained by the decrease of the relative signal-to-noise ratio during the distillation process compared to distilling from 256-(Meng et al., 2023) or 1024(Salimans & Ho, 2022) DDIM steps.

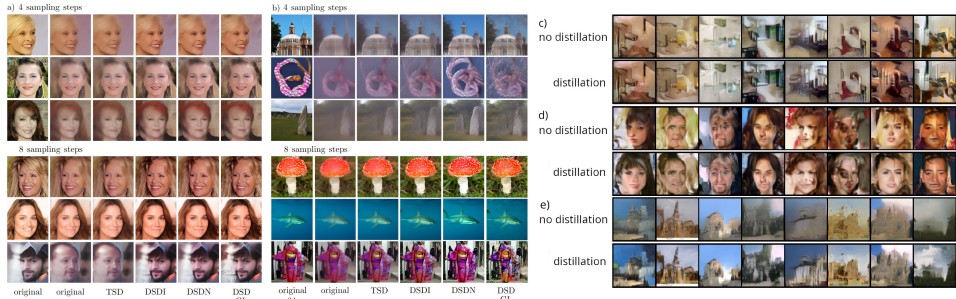

Figure 2: (left) Random $a$), **4- and 8-step samples** from the distilled unconditional CelebA-HQ 256x256 model and $b$) the distilled conditional ImageNet-256$^2$ model, including 64-step samples from the undistilled models. (right) Random **128-step** samples the from trained and trained-distilled models for $c$) LSUN Bedroom, $d$) CelebA, and $e$) LSUN Church Outdoor datasets.

## 5.2 Training DSD

We apply training DSD on three separate datasets, the 64x64 pixel versions of the CelebA, LSUN Bedroom and LSUN Church Outdoor datasets. The loss function $L$ is defined as a weighted sum of the training loss and the distillation loss.

All three models were trained with $v$-prediction as their output, similar to that described in Salimans & Ho (2022). For celebA, a total of 65 000 parameter updates were performed at a learning rate of $10^{-4}$, whereas the LSUN Bedrooms model was trained for 50 000 parameter updates at a learning rate of $4 \cdot 10^{-4}$ with the LSUN Church Outdoor model being trained at the same learning rate for just 40 000 parameter updates. For distillation during training, $\alpha$ and $\beta$ were set to $[0.85, 0.15]$, respectively. Setting $\beta$ to a similarly low value inhibits the affect of the distillation loss when the model has not matured yet during the early stages of training. During this phase, consecutive DDIM steps still show little variation, lowering the importance of distillation at this time. In the later stages when training loss stabilizes and successive DDIM steps add improvements to image fidelity, distillation loss becomes more important as the training loss stabilizes. Further research could attempt a dynamic weight scheduling between these two parameters to optimize distillation during training.

Post-training samples for all three models are shown in 2, with the corresponding FID scores depicted in 3. For the LSUN Bedroom model, subtle improvements to features in the images can be seen when distillation is performed, with the final FID score improving mostly towards the end of the training process, during which the distillation loss affects the model's weight updates the most and we see that training without distillation start to plateau. However, such improvements were not clearly seen for the CelebA dataset, where despite subjective improvements between the samples with and without distillation, no clear improvements to FID is found. Interestingly, FID scores show a nearly identical trend throughout the training process. The early stabilisation of the FID scores around just 20 to 25 thousand steps may be indicating that the train loss decreased more rapidly

than was the case for the LSUN Bedrooms dataset, where the distillation loss had relatively little impact later on during training. For the the LSUN Church Outdoor dataset, a more gradual decline in FID is seen, with the distilled model finally overtaking during the later training phase. Similar to the results of the LSUN Bedroom model, distinct image features seem to be present in the distilled model before yet visible in the non-distilled counterpart.

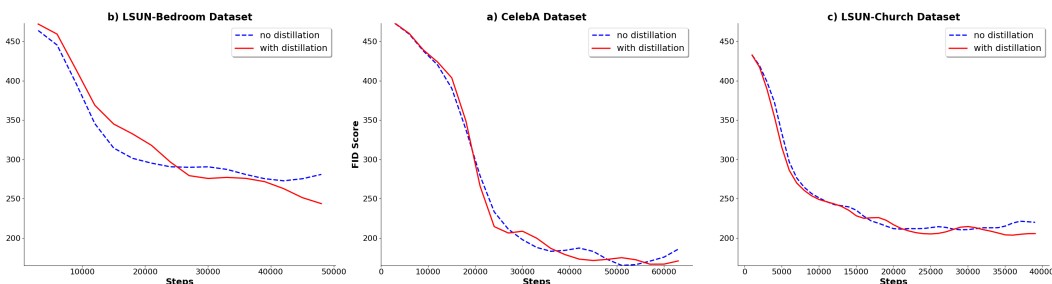

Figure 3: FID scores for the trained and train-distilled models on the **a)** LSUN Bedroom, **b)** CelebA, and **c)** LSUN Church Outdoor datasets.

## 6 CONCLUSION

Our results demonstrate the applicability of self-distillation to diffusion settings. Both for fine-tuning a pre-trained model, and for training a diffusion model from scratch.

We have also tested different schedules for reducing the sampling steps during (self-) distillation. We find that on the datasets tested, a linear reduction may perform better than the halving schedule used in progressive distillation, but the specifics appear highly dependent on the task.

In some cases, a naive approach without any scheduling (DSDN) performs best. This could be explained by the relative increase in parameter updates during lower depths compared to other DSD approaches, similar to the Progressive Distillation techniques where the total number of denoising steps performed is lowered after each distillation step.

By forgoing the use of a separate teacher model, DSD reduces the network evaluations required per update of the parameters from 3 (Ho & Salimans) down to just 2. This also implies that, unless the teacher outputs have previously been cached, DSD frees up the GPU memory taken up by a teacher model, reducing the hardware requirements of diffusion model distillation. DSD also demonstrates improved convergence during distillation, requiring fewer total parameter updates for reaching a similar or improved level of image quality.

**Limitations and future work** One important limitation of DSD is the requirement to train sequentially, in concert with the denoising process. This means that the samples that we train on are not fully i.i.d., but in practice this does not appear to seriously impact learning. The benefit is that no dataset is required during fine-tuning, simplifying the training process.

Our results on training DSD suggest that self-distillation may be employed favorably in training models from scratch. However, how this approach behaves at scale, and whether it leads directly to models that generate high-quality images in 1-10 samples, without a two-step training process remains to be seen. Additionally, studying the combination of DSD with existing non-direct distillation approaches might demonstrate DSD's strengths in quickly decreasing the required number of sampling steps before distilling down to 1-2 sampling steps using existing methods. Answering this question experimentally likely requires substantial experimental resources. We leave this question to future work.

# 7 REPRODUCIBILITY STATEMENT

Our code is publicly available at `https://anonymous.4open.science/r/DSD`. The hyperparameters used for the DSD implementations are provided in Table 3 in the Appendix. The full specifications for the models used in training DSD are available as configuration files in the repository. The processes for calculating the FID and Inception Score metrics for all models are likewise made available. All distilled model checkpoints, including all generated samples for each implementation and number of DDIM steps, will be made available on the public repository.

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

## H  HYPERPARAMETERS

| | Params (M) | Updates | Scheduling | Lr-start | Lr-end | Adam $\beta_1$ | Adam $\beta_2$ | EMA decay | Grad. clip |
|---|---|---|---|---|---|---|---|---|---|
| ImageNet-$256^2$ | 400.92 | 5000 | cos. ann. | $1 \cdot 10^{-7}$ | $1 \cdot 10^{-8}$ | 0.99 | 0.999 | 0.9995 | 0-1 |
| CelebA-HQ | 274.06 | 4000 | cos. ann. | $1 \cdot 10^{-6}$ | $1 \cdot 10^{-7}$ | 0.9 | 0.98 | 0.995 | 0-1 |
| LSUN Bedroom | 274.06 | 4000 | cos. ann. | $1 \cdot 10^{-6}$ | $1 \cdot 10^{-7}$ | 0.9 | 0.98 | 0.995 | 0-1 |

Table 3: Hyperparameters used for the Direct Self-Distillation attempts. All parameters were kept identical between the different update schedules performed.

## I  DSD SCHEDULING

Figure 4 shows the relative importance of each DDIM step during TSD/DSDI, DSDN, and DSDGL. Given the nature of DSD in which, during the regular sampling procedure, successive sampling steps are compared against each other, changing the total steps $N_i$ at any point will alter the distribution of different sampling steps seen. This is particularly apparent in TSD/DSDI, where halving of the total $N_i$ takes place at regular intervals indicated by a sharp drop in the frequency curve. Figure 4b shows this same principle along with the added weighting of the cosine annealing learning rate during the entire process of distillation. Given the learning rate's warm-up, followed by a decrease over time, the relative strength of updates at specific intervals of the sampling process can be significantly altered, as is especially apparent for TSD/DSDI.

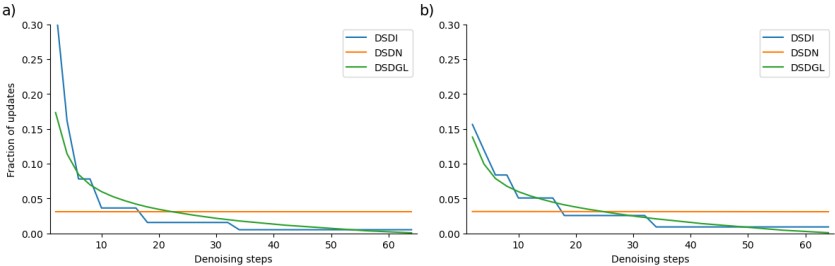

Figure 4: Comparison of the allocation of parameter updates at various step-sizes for all distillation procedures with *a)*, indication of the proportion each step size has been witnessed during distillation and *b)*, the same proportion but scaled by the cosine annealing learning rate, effectively the proportion of influence on distillation per DDIM step.

## J  EXPONENTIAL DIRECT SELF-DISTILLATION

$$U_i = \frac{e^{(S^{-1})_{n-1}}}{\sum_{j=1}^{n} e^{S_j}} \cdot P, \text{ where } S = \{1, 2, 4, 6 .. N\}, \text{ and } P \subseteq \mathbb{R} \tag{2}$$

In order to explore the performance of Direct Self-Distillation when the model's parameter updates are performed mainly at fewer DDIM steps, we also propose an **exponential update schedule (DSDGEXP)**, aiming to gradually increase the number of parameter updates as the model distills towards fewer DDIM steps, iteratively lowering the total steps performed. Although this approach seems to put emphasis on early image feature introduction, subjective analysis of image quality presented in Figure 5 seems to indicate that DSDGEXP fails to retain overall smoothness, showing what can be described as over-sharpening of the final denoised images. Potentially, this is the result of the model being distilled primarily at low DDIM steps with low SNR ratios. The model might therefore learn to remove large amounts of noise at each timestep, losing its ability to reconstruct finer image details. FID and IS scores of the conditional ImageNet256 model and FID scores of the unconditional CelebAHQ and LSUN Bedroom models are detailed in Table 4.

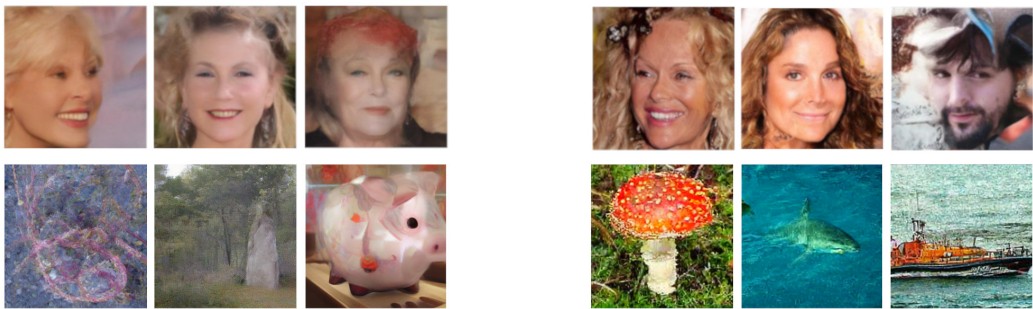

Figure 5: Random **4-step** (left) and **8-step** (right) samples from the DSDGEXP distilled conditional ImageNet-256$^2$ (bottom) and unconditional CelebA-HQ 256x256 (top) models.

| | FID($\downarrow$)/IS($\uparrow$) | | |
| | S=2 | S=4 | S=8 |
|---|---|---|---|
| ImageNet-256 | 132.39/4.03 | 24.25/42.88 | 31.71/43.70 |
| CelebA-HQ | 237.67/- | 103.28/- | 79.48/- |
| LSUN Bedroom | 266.86 /- | 88.66 /- | 27.76 /- |

Table 4: DSDGEXP FID scores for the distilled CelebA-HQ and LSUN Bedroom models and FID/ IS scores for the distilled conditional ImageNet-256$^2$ model at *2, 4,* and *8* DDIM steps, calculated over 30K sample images per model/step.

## K  SAMPLES

### K.1  CELEBA-HQ-256$^2$

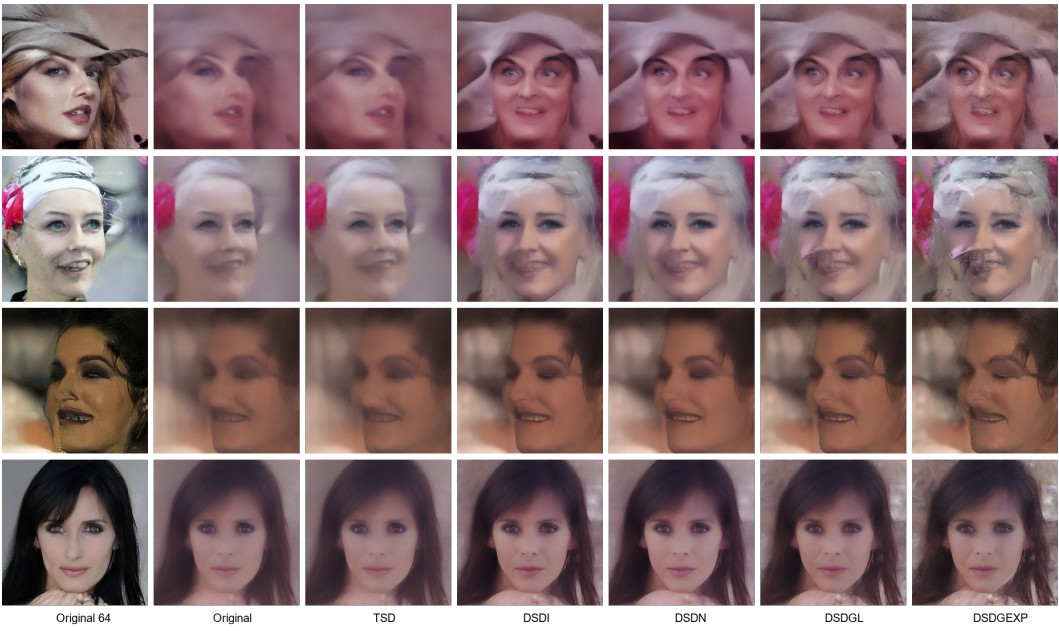

Figure 6: Random **4-step** samples from the original and distilled unconditional CelebA-HQ 256x256 models. The *original 64* row contains the 64 sampling step reference images generated by the original model.

### K.2  IMAGENET-256$^2$

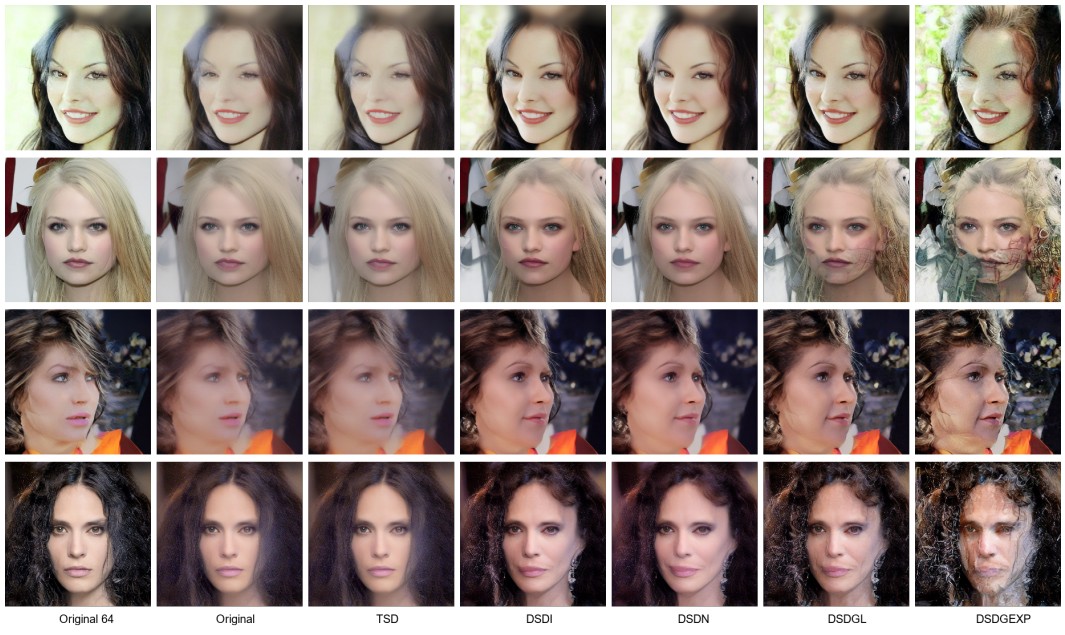

Figure 7: Random **8-step** samples from the original and distilled unconditional CelebA-HQ 256x256 models. The *original 64* row contains the 64 sampling step reference images generated by the original model.

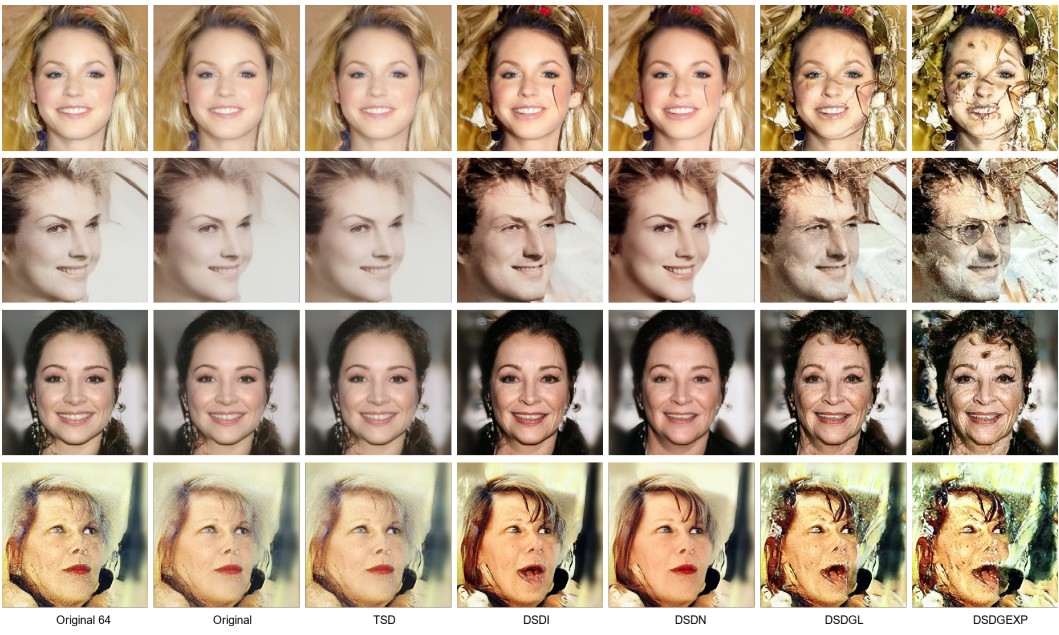

Figure 8: Random **16-step** samples from the original and distilled conditional CelebA-HQ 256x256 models. The *original 64* row contains the 64 sampling step reference images generated by the original model.

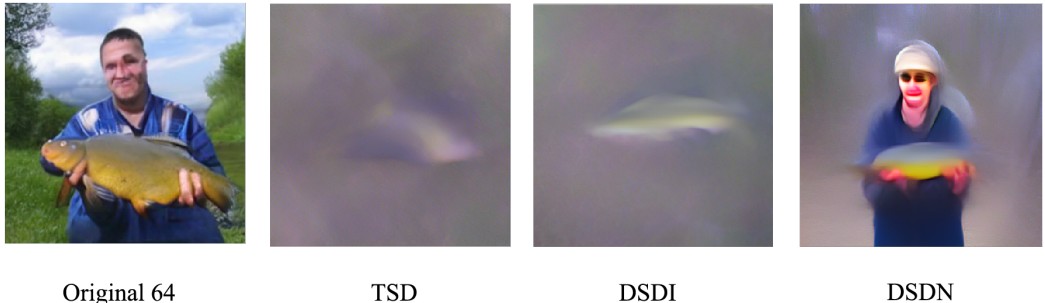

| Original 64 | TSD | DSDI | DSDN |

Figure 9: A close-up comparison between **2-step** *'tench, Tinca tinca'* samples from the original and distilled conditional ImageNet 256x256 models. The *original 64* row contains the 64 sampling step reference images generated by the original model.

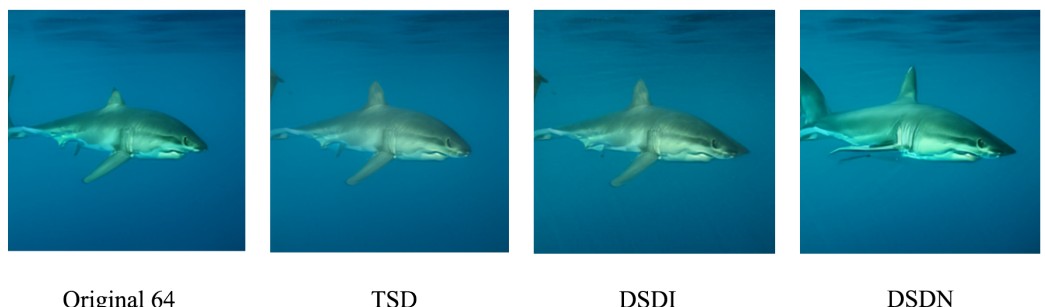

| Original 64 | TSD | DSDI | DSDN |

Figure 10: A close-up comparison between **8-step** *'great white'* samples from the original and distilled conditional ImageNet 256x256 models. The *original 64* row contains the 64 sampling step reference images generated by the original model.

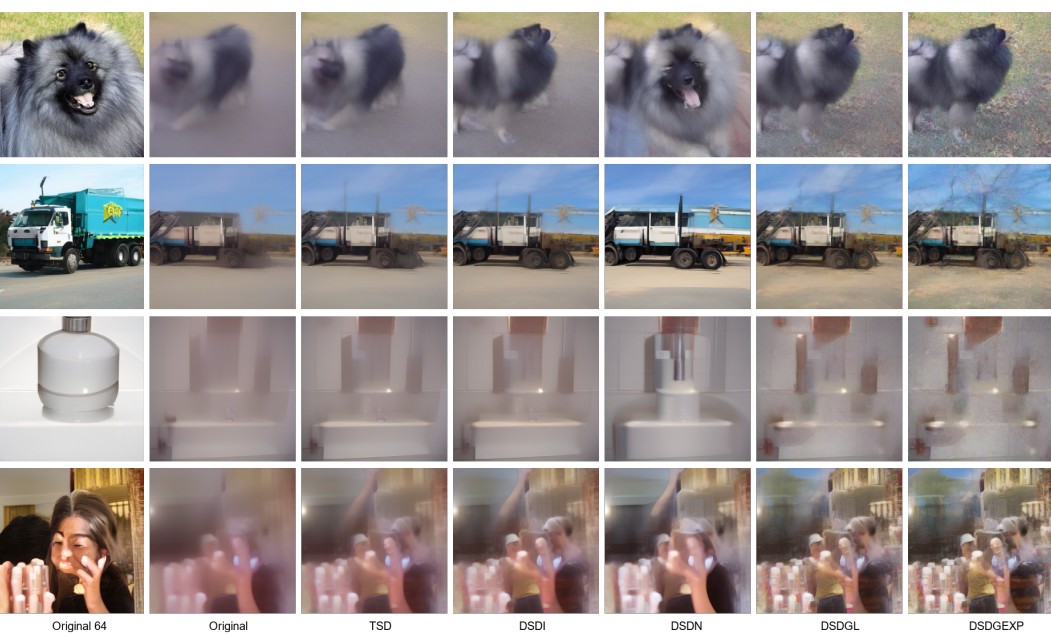

Original 64    Original    TSD    DSDI    DSDN    DSDGL    DSDGEXP

Figure 11: Random **4-step** samples from the original and distilled conditional ImageNet 256x256 models. The *original 64* row contains the 64 sampling step reference images generated by the original model.

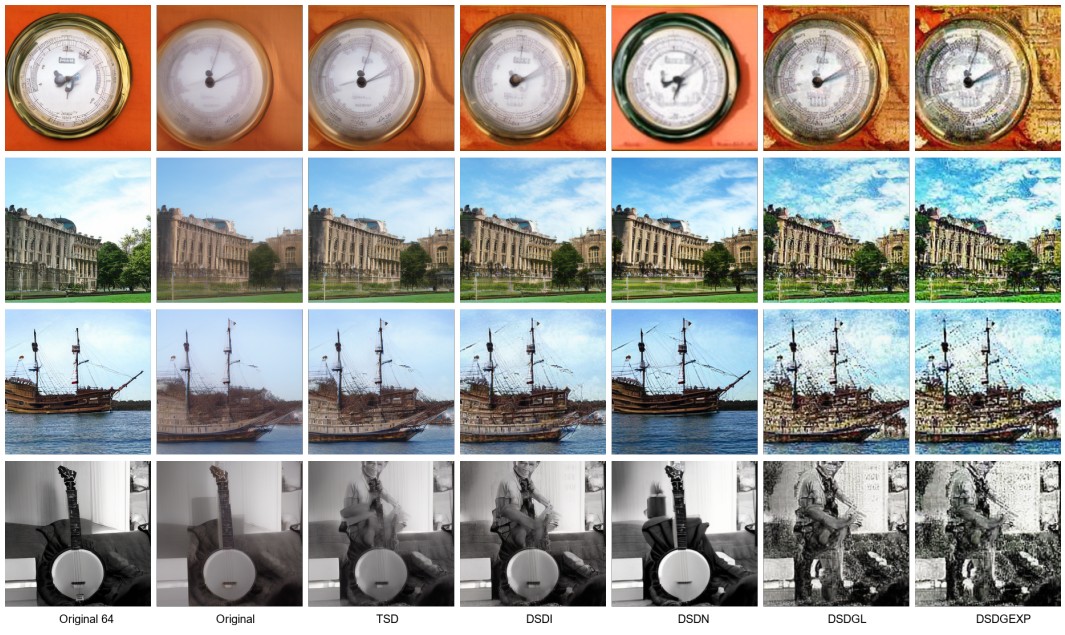

Figure 12: Random **8-step** samples from the original and distilled conditional ImageNet 256x256 models. The *original 64* row contains the 64 sampling step reference images generated by the original model.

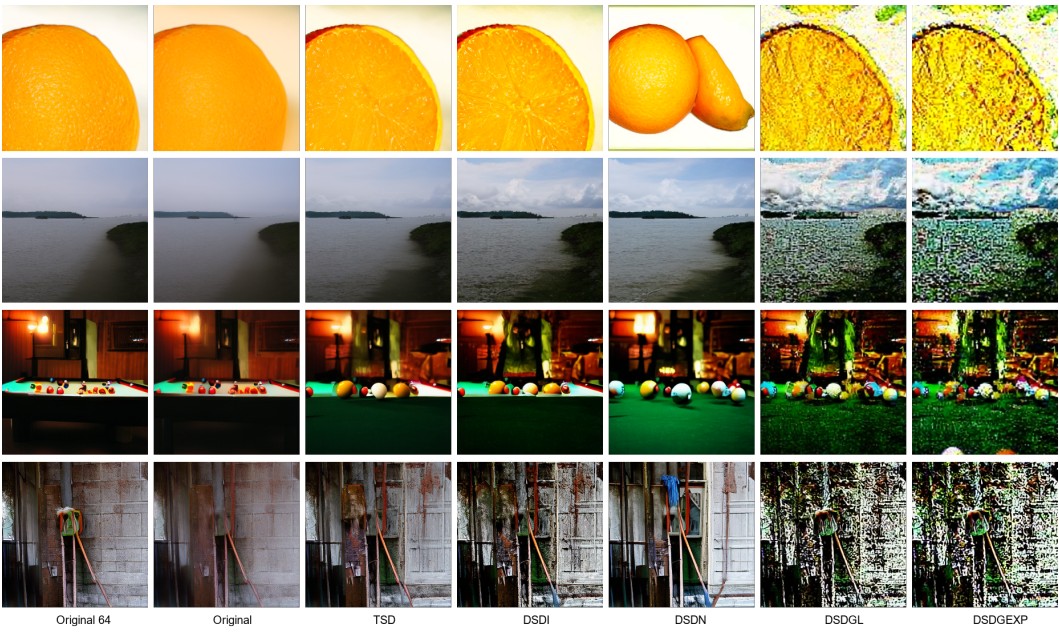

Figure 13: Random **16-step** samples from the original and distilled unconditional ImageNet 256x256 models. The *original 64* row contains the 64 sampling step reference images generated by the original model.

