# OpenReview forum: "Self-distillation for diffusion models"
_ICLR.cc/2024/Conference — Submitted to ICLR 2024_

### Official Review · Reviewer_Robz · 2023-10-31

**Soundness:** 2 fair
**Presentation:** 2 fair
**Contribution:** 1 poor
**Rating:** 3
**Confidence:** 4

**Summary:**

This work proposes direct self distillation  (DSD) which is a method to distill diffusion models into itself. The core idea is to perform two consecutive sampling steps to get the latents $z_t, z_{t-1}$ and $z_{t-2}$, and use $x_\theta (z_{t-1})$ as prediction and  $x_\theta (z_{t-2})$ as target and minimize the squared distance between them.

The primary distinction from prior works on online diffusion like Progressive distillation (PD) is that PD needs 3 network evaluations per parameter update of student model while DSD needs 2 network evaluations per parameter update. It is worth noting that offline distillation methods also use fewer network evaluations (usually 1 or 2 evaluations) than online distillation methods.

Overall this work does not aim to distill down diffusion models to as few sampling steps as possible but rather to compare DSD to PD while reducing the required model updates for similar image quality.

**Strengths:**

1. The proposed method reduces GPU compute cost as it reduces the number of network evaluation per parameter update from three to two. Self distillation eliminates the need to maintain 2 copies of the model in memory (for student and teacher), in turn reducing overall compute requirements.
2. DSD also uses lower number of parameter updates (4K-5K) compared to prior online distillation techniques which need thousands of parameter updates. For instance, results in Table 2 for CelebA HQ and LSUN Bedroom with 8 steps uses 4K parameter updates performs better than teacher-student distillation at similar number of parameter updates.

**Weaknesses:**

1. Advantages of self-distillation while training diffusion models from scratch is unclear and additional results are needed to demonstrate its benefits while training from scratch. Figure 3 compares FID scores for distilled and the regular diffusion model while training for scratch only for the first 30k-60K steps. These FID scores are quite high for DDIM and longer training is needed. For instance, on CelebA, the original DDIM (with 10 sampling steps) achieves FID of 17.33 while in Figure 3, the lowest FID is around 100. Thus longer training will help improve FID scores, however It is unclear whether the gains seen in these plots due to self-distillation will continue for the entire duration of training, and whether the final FID scores of model that is trained from scratch with self-distillation is strictly better than the model that is trained without any self-distillation.

2. I understand that the primary motivation of DSD is to use fewer parameter updates while achieving comparable image quality with progressive distillation, however the current FID scores in Table 2 are high across the datasets. Further, DSD does not result in high quality images, in terms of FID, in few step sampling (2-4 steps). In order for someone to strictly prefer DSD over PD, one would have to show at least one of the two:  1) DSD can achieve comparable or better FID than PD while using fewer parameter updates, incase we use fine-tuning DSD. 2) Using DSD while training from scratch simultaneously improves FID and reduces sampling steps than training a model from scratch and later doing PD. Currently, the paper does not include sufficient results to show either 1) or 2). DSD seems to outperform TSD at fewer parameter updates (From Table 2) but it is unclear if DSD can match FID of PD with more parameter updates. Currently, the models in the paper use 4K-5K parameter updates. Further, DSD quickly deteriorates in terms of FID while using 2 and 4 step sampling. The results would be much more convincing if FID scores can be matched with PD with slightly more parameter updates. A table could be added that shows FID, sampling steps, and NFEs for DSD and PD. Ideally, we should have comparable FID and sampling steps with much fewer NFEs than PD. It is also worth noting that DSD is orthogonal to PD. Thus it should be possible to use DSD, with say 64 sampling steps which might have better FID, and then do PD to get distilled model with better FID scores at 2-4 sampling steps.

**Questions:**

1. Does DSD maintain an exponentially moving average (EMA) copy of the model weights during training/fine-tuning?
2. Can you help me understand Figure 2? Specifically, I don’t understand X axis. My understanding is that this is $N_i$. Why are the number of denoising steps along X axis small? Shouldn’t we have large step sizes $N_i$ in self distillation initially?
3. I am not sure that this statement in introduction (second paragraph) about the current distillation techniques is true — “The student is trained to mimic the teacher, but in fewer iterations, so that, eventually the teacher and the student diverge during training.” In my experience with distillation techniques, I haven’t seen that student and teacher diverge in the cases when distillation is successful. Could the authors elaborate on what they mean by “diverge” here?

Minor Suggestions:
1. The last line in abstract has a minor grammatical error. It currently reads “help the model to convergence produce high-quality samples more quickly.” This sentence could be revised to “help the model converge quickly, ultimately producing higher-quality samples."
2. The readability of paper would greatly improve if the notation is made more consistent.
    - The loss in Eq. 1 uses the notation of $\alpha \cdot \||\hat{z}\_{t-1} - z_{t-1}\||^2 + \beta \cdot \|| \hat{z}\_{t-1} - \hat{z}\_{t-2}||^2$ but previous sections use $\||\hat{x}\_{t-1} - \hat{x}\_{t-2}\||^2$ as loss for distillation. Further, the usual choice of loss objective for training diffusion models is $\||\hat{x}\_{t-1} - x_{gt}\||^2$ ($x_{gt}$ is ground truth image) or $\||\hat{\epsilon}\_{t-1} - \epsilon\||^2$. Why was  $\||\hat{z}\_{t-1} - z\_{t-1}\||^2$ chosen instead for training (I'm not asking about the second loss term that corresponds to self distillation loss here)?
    - Minor: The results section uses S (See header for Table 1) without defining it in previous sections. I presume this denotes number of sampling steps in distillation.

---

> ### Comment · Reviewer_Robz · 2023-11-22
>
> I'm updating my score as my concerns haven't been addressed so far.

---

> ### Author Response · Authors · 2023-11-22
> **Pre-revision comment**
>
> Dear reviewer,
>
> We are currently still revising some minor aspects of our paper, which we wanted to submit alongside the comments we had already prepared. However, given your additional comment this morning, I would like to instead respond to your remarks before uploading our final revised paper. Alongside many minor improvements to readability and structure, these are the main changes based on your review (which have been highlighted in the document):
>
> * Your points regarding the high FID scores of distillation during training are justified. Given the comparatively pilot-like nature of DSD during training, previous FID scores were calculated with a relatively low number of samples generated for score calculation. We have re-run the experiments (with more epochs) and used our time since the last submission to optimize the hyperparameters for the training. Given the improved convergence of training overall, our latest results now demonstrate more clearly the effect of additional direct self-distillation during the standard process.
>
> * We feel that our relatively underwhelming scores for the DSD applications were at least in part due to the increase in generated images for calculation. For example, our imageNet results were calculated over 30K samples, against 5K of the existing PD variant. Lowering our samples to match the 5K shows Inception Scores similar to those of PD. We have therefore opted to instead use the evaluation code provided by the baseline models’ papers, along with the decision to calculate scores over 5K samples. Current scores now lie within the vicinity of those provided by the baseline papers.
>
> * If the computation time allows for it, we will also attempt longer DSD runs to try and match the quality found at fewer sampling steps. We are currently limited in our ability to fine-tune the DSD process, which in part explains our focus on showing the working principle to be efficient and simple, rather than directly aiming to compete with matured existing methods. On a side-note, your idea of having DSD lower the sampling steps initially before introducing PD is one we have considered as well. We feel this idea is perhaps too much to introduce at this stage of the revision, but we will include it in our ideas for future work.
>
>  * We do indeed keep the EMA weights during distillation and train-distillation. Sampling for FID/IS calculation, as well as the samples shown in the paper, have been done with the EMA weights loaded.
>
> * Regarding Figure 2, we have improved the explanation of the graph, which has received a dedicated section in the Appendix. This graph essentially shows during the entire distillation process, how often a specific $x^{’}(z_t)$ (for instance, the 3rd sampling step) was distilled into $x^{‘’}(z_{t-1})$ (then the 4th sampling step, in reverse order). For this reason, the x-axis spans 0-63. A conceptually simple example is DSDN, which loops from step 0-63 and updates each step against the consecutive sampling step’s output. Therefore, at each sampling step, an equal number of parameter updates has taken place. If we were to halve the total sampling steps at some point as done in DSDI, higher sampling steps (>32) are no longer seen, lowering their respective y-axis frequencies. Please let us know if this clarifies the graph somewhat.
>
> * We agree that this was confusingly worded. What this passage refers to as divergence, is the effect that one iteration of the student model, produces approximately, the same effect as several iterations of the teacher model. In short, distillation still aims to have the student mimic the teacher, but because the former uses fewer iterations the behavior of one iteration of the UNet diverges between student and teacher. We have clarified this in the manuscript as follows:
>
>   "The student is trained to mimic the teacher, but in fewer iterations, so that in time the student converges to replicate the teacher’s sample quality at fewer sampling steps"
>
> * Finally, we have included your improvements regarding notation.
>
> Other notable improvements:
>
> * Clear definition of the role of TSD
> * Dedicated introduction of PD
> * Erroneous notation in Equation 1. has been addressed
>
> Thank you for your time and patience. Your comments have proven very valuable during the revision process.

---

> > ### Comment · Reviewer_Robz · 2023-11-22
> >
> > Dear authors,
> >
> > Thank you for detailed comments to my questions and for adding in the new results. The updated writing also makes the central ideas of the paper more clear. I believe that the paper will benefit if following additional analyses and experiments are added:
> > 1. The current distillation results are still unconvincing. It would be really useful if lower FID scores can be achieved, even if it comes at the cost of more parameter updates.
> > 2. Table 2 should specify whether the distillation was achieved from a pretrained model, or via training from scratch. Ideally, there should be two separate tables that indicate efficiency of DSD for both the cases, and compares it against PD.
> > 3. It would be useful to add in recall metric to indicate if modes are being dropped due to fewer parameter updates.
> > 4. Table 1 indicates values of FID/IS with DDIM sampling for fewer steps. I would argue that the goal of distillation is to match the performance of longer diffusion chains in a few steps, Therefore, comparing against undistilled model at fewer steps is not completely fair.
> >
> > Overall, I do think that the motivation for this paper is sound but additional experimental results are necessary to make a convincing case for DSD.

---

> > > ### Author Response · Authors · 2023-11-22
> > > **Response**
> > >
> > > Dear reviewer Robz,
> > >
> > > * I will attempt some other FID calculations. I suspect that our generation of the .NPZ files might be hindering the comparison against the original dataset images, given the strong results in terms of IS scores.
> > > * Table 2 represents only the results of the distillation of pre-trained models, not the results of distillation during training. To compare our distillation during training against any PD implementations would likely require a computational budget beyond our scope, given their resolutions of $256^2$ compared to our train-distilled $64^2$. However, implementing DSD in combination with PD to potentially achieve high quality, low DDIM samples more efficiently, is an approach our computational budget would allow as future research.
> > > * I will see about the recall metric, I understand your concerns.
> > > * If I understand you correctly, you argue that comparing results from a distilled model requires a sort of *looking up* at the undistilled model's quality further along the sampling chain. I will add these comparisons in our final submissions. For instance, how the distilled ImageNet 8-step model performs on-par with undistilled 32- and (close to) 64 step models.
> > >
> > > I'm glad to hear you find the paper to be improved after our latest revision. Hopefully, I will also be able to incorporate these improvements before the deadline.

---

> ### Author Response · Authors · 2023-11-23
> **FID scores**
>
> Dear reviewer Robz,
>
> I'm currently in the process of fully re-generating all .NPZ files for use with the evaluation script mentioned before. It is clear now that my presumption of falsely generated .NPZ files is most likely correct, as I see drastically improved FID scores. Luckily, I have a local fast-storage backup of all our images, which will hopefully allow me to update all scores before the deadline. Thank you for your patience once more.
>
> EDIT: I have updated the distillation scores, highlighted in Tables 1 and 2, as well as indicated these to be results from the distillation of pre-trained models, unlike distillation during training.

---

### Official Review · Reviewer_EnYo · 2023-11-05

**Soundness:** 2 fair
**Presentation:** 1 poor
**Contribution:** 2 fair
**Rating:** 3
**Confidence:** 4

**Summary:**

The paper introduces a self-distillation methodology that improves Progressive Distillation (PD). This approach is applied to distill diffusion models to generate high-quality samples more quickly. It diverges from previous methods like PD that utilize a static teacher model by integrating both teacher and student roles within the diffusion models. Furthermore, it avoids accessing the training dataset, initiating instead from a noise distribution. Experiments demonstrate that this distillation method reduces the number of model evaluations required for image synthesis.

**Strengths:**

- The self-distillation approach removes the static teacher from distillation and simplifies the distillation process from PD.
- The proposed method circumvents accessing the training data. This allows for data-free distillation.
- This approach converges faster than PD and achieves comparable image quality using a restricted number of function evaluations (NFEs).

**Weaknesses:**

The manuscript is not prepared for acceptance due to several substantive concerns.

1. Clarity of writing. The manuscript suffers from significant issues with clarity, which impedes the reader's understanding of the methodological details. For example, on page 2, the latent variables  $\mathbf{z}_t$ are introduced without adequate context, leading to confusion. It is only in Section 3 that the reader learns these notations are adopted from Progressive Distillation (PD). Furthermore, the manuscript lacks a necessary introduction to PD and fails to clearly articulate the distinctions between Teacher-Student Distillation (TSD) and PD. The writing quality would benefit from thorough proofreading and reorganization.

2. The complexity of the method. The proposed method entails a three-level nested loop, as presented in Algorithm 2. Although it is not technically unacceptable, without a clear introduction of the notations, the writing issue complicates the understanding of the algorithm and scheduling. The recycling of previous latent variables in the loop, coupled with the scheduling, together add to the complexity. Despite eliminating the static teacher, the method does not appear to be a conceptual simplification of PD.

3. Inadequate experiment comparisons. The popular benchmarks for this task, for example, CIFAR-10 and ImageNet 64$\times$64, are missing from the current manuscript. The experiment's design makes it unable to compare with recent progress. Furthermore, the algorithm seems not to demonstrate clear superiority over TSD or the original models in some cases.

4. Missing literature. Distillation methods developed subsequent to PD are missing in the manuscript. There is a notable absence of current data-free distillation methods and those that have transitioned from using a static teacher to an online teacher. Progress on diffusion samplers without retraining should be introduced in the literature.

**Questions:**

Please see the comments above.

---

> ### Author Response · Authors · 2023-11-22
> **Our Revisions**
>
> Dear reviewer, thank you for your time addressing the concerns you have found. Please allow me to address these points, in no particular order:
>
> * The DSD algorithm's depiction is indeed over-complicated beyond necessity, where we aimed to include every possible scenario of the different DSD implementations (and their respective update schedules) into a single formulaic representation. Instead, we have removed those parts trying to encapsulate our step-scheduling, making the DSD algorithm instead a true, generalized application of the concept.
>
> * We will also aim to simplify the document in that the concept of PD naturally flows into our approach, which indeed is not the case currently. We will also try to reformulate TSD in an attempt to better distinguish this approach from both DSD and PD, since TSD is neither Direct, or dataset–driven like PD.
>
> * The Inception Scores for the conditional models have also been updated, which are now calculated using the evaluation script provided by the authors of the pre-trained models. These now seem to correctly highlight image quality post-distillation.
>
> Thank you for your time and patience. We have submitted our revised work, with non-trivial changes highlighted for comparison.

---

### Official Review · Reviewer_tQwJ · 2023-11-06

**Soundness:** 2 fair
**Presentation:** 3 good
**Contribution:** 2 fair
**Rating:** 3
**Confidence:** 4

**Summary:**

This paper proposes a Direct Self-Distillation (DSD) method for diffusion models. DSD allows the model to distill itself during training, resulting in outputs in fewer iterations. Based on the results presented in the paper, DSD appears to improve the convergence rate of training. However, in the context of distillation models, the goal is typically to train a student model that achieves similar performance to the teacher model but with fewer resources. Unfortunately, the experiments of this paper show that the distilled model has significantly worse performance than the teacher model, which makes the reader highly suspect the validity of the method.

Additionally, it is important to compare the number of parameters in the distilled model to the original model. This information would demonstrate the potential benefits of the distilled model in terms of model size and resource requirements.

**Strengths:**

1. The problem of distilling diffusion models to other efficient models is important for efficient sampling of diffusion models.

**Weaknesses:**

1. The most severe weakness of this paper is the weak performance. It is well-known that the goal of distillation is to obtain more efficient student models with comparable performance to teacher models. However, in this paper, the proposed distilled model is not even close to the teacher model (LSUN results in Table 2). I strongly recommend the author to check the results and validity before submitting.

2. I recommend the author compare the number of parameters in the distilled model to the original model. This information would demonstrate the potential benefits of the distilled model in terms of model size and resource requirements.

3. Some minor points such as writing and presentation.
(1) The 4th line of related work sections should use citations with brackets;
(2) I recommend the author combine Figure 3 and Figure 4 into one line such that a lot of space can be saved.

**Questions:**

See weakness

---

> ### Author Response · Authors · 2023-11-22
> **Our revisions**
>
> Dear Reviewer, thank you for your insights.
>
> From your assessment it seems clear that our work would benefit from some improvements regarding clarity of results. To address your point regarding the number of parameters of Teacher vs Distilled models, the DSD distilled-models are exact copies of the teacher model in terms of structure and number of parameters. Given the nature of DSD, it might in fact be in error to state that there's a teacher model, since for all DSD approaches, this is in fact the distilled model before the first parameter update (only a single model is ever sampled from). We have clarified this in the latest version of our paper.
>
> In similar need of clarification is the definition of our baseline models. I believe that, given your reference to Table 2, you may have interpreted the "PD" column as being our teacher model. Instead, our teacher models are displayed in Table 1. The "PD" column refers to the distillation results of other, non-DSD approaches, of which the checkpoints are sadly not available to us.
>
> The great leap in image quality of these teacher/original models compared to ours was touched upon in the Results section, but it has come to our attention that our implementation of IS calculation was rather non-forgiving, as our decision to include many more samples for score-calculation increased robustness at a significant loss of maximum scores. Instead, we have now opted to use the models’ original authors’ evaluation code, leading to scores more in line with perceived image quality overall. For instance, our ImageNet results show higher Inception scores for 8 steps compared to PD (~140-200 vs 128.63), with DSDN narrowly managing to show improvements at just 4 steps (126.61 vs 126.36). We’ve also matched the PD papers’ choice of calculating the scores over 5000 samples instead.
>
> Your other points regarding the writing/presentation are also very welcome and have been implemented in the latest revision. Large changes, excluding any minor improvements to readability, have been highlighted in our final document, which we feel improves the clarity of our results.
>
> Thank you very much for your time.

---

### Meta-Review · Area_Chair_efnd · 2023-12-17

**Metareview:**

The reviewers unanimously supported rejecting the paper, as do I. The manuscript was confusing in presentation, the problem was of questionable value, and the method didn't seem to perform well.

**Justification For Why Not Higher Score:**

No reviewers made any case for acceptance.

**Justification For Why Not Lower Score:**

N/A

---

### Decision · Program_Chairs · 2024-01-16

Reject